# Synthesis and Electromagnetic Interference Shielding Performance of Ti_3_SiC_2_-Based Ceramics Fabricated by Liquid Silicon Infiltration

**DOI:** 10.3390/ma13020328

**Published:** 2020-01-10

**Authors:** Xiaomeng Fan, Yuzhao Ma, Xiaolin Dang, Yanzhi Cai

**Affiliations:** 1Science and Technology on Thermostructural Composite Materials Laboratory, Northwestern Polytechnical University, Xi’an 710072, China; yuzhaoma@126.com (Y.M.); dangxiaolin@nwpu.edu.cn (X.D.); 2College of Materials Science and Engineering, Xi’an University of Architecture and Technology, Xi’an 710055, China; caiyanzhi@xauat.edu.cn

**Keywords:** Ti_3_SiC_2_, liquid silicon infiltration (LSI), electromagnetic interference, shielding effectiveness

## Abstract

In this work, Ti_3_SiC_2_-based ceramics were fabricated by the infiltration of liquid silicon into TiC preform by incorporating a small amount of Al. Al can play a catalytic role to promote the formation of TiC twins before liquid silicon infiltration (LSI), which leads to the increase of transformation efficiency from TiC to Ti_3_SiC_2_ in the LSI process. When the Al content in the TiC preform increases to 9 wt.%, the volume content of Ti_3_SiC_2_ reaches 85 vol.%, revealing the high electromagnetic interference shielding effectiveness of 39 dB in the frequency range of 8.2–12.4 GHz. The results indicate that it is an effective way to synthesize Ti_3_SiC_2_-based ceramics with excellent electromagnetic shielding performance.

## 1. Introduction

Electromagnetic interference (EMI) shielding materials have attracted more attention due to their extensive applications in protecting electronic devices from electromagnetic interference [1,2,3]. Metal-based and high-conducting polymer-based composites were the two most common types of EMI shielding materials. However, the metal is easily corroded, and the polymer cannot be applied at high temperatures. It is well known that ceramics have low density and high corrosion resistance, which can be applied as high-temperature structural materials, but the low electrical conductivity limits their application as EMI shielding materials. Apart from conventional ceramics, MAX phases have high electrical conductivity-like metals, due to the existence of M-X metallic bonding in the lattice structure, which reveals their superior application potential as EMI shielding materials [4,5,6,7,8,9].

Bulk Ti_3_SiC_2_ exhibited high complex permittivity, and the EMI shielding effectiveness (SE) reached 35–54 dB in the frequency range of 8.2–18 GHz (X-band and Ku-band) [4]. With the addition of Ti_3_SiC_2_ filler, the EMI SE of polyaniline composite was greatly enhanced [5]. Bulk Ti_3_AlC_2_ with a high texture degree had an EMI SE of above 30 dB from room temperature to 800 °C [6]. MAX phases modified ceramic matrix composites were also prepared, and their EMI SE can be higher than 30 dB in the frequency range of 8.2–12.4 GHz [10,11,12,13]. MXenes, as a new family of two-dimensional materials, exhibited outstanding EMI performance with the SE value being higher than 90 dB [14].

Ti_3_SiC_2_, as one of most studied MAX phases, can be prepared by several methods such as hot pressing [15], spark plasma sintering (SPS) [16], and reactive melt infiltration (RMI) [17,18,19,20,21]. Among these methods, RMI is an effective way to synthesize dense Ti_3_SiC_2_-based ceramics with near-net-shape. In the RMI process, metal melt (Si/Al-Si) infiltrated into porous TiC preforms driven by capillary force, and then reacted with TiC particles to synthesize the dense Ti_3_SiC_2_-based ceramics [18,19]. Carbon was added into TiC preform to promote the precipitation of Ti_3_SiC_2_ in the liquid silicon infiltration (LSI) process, and the volume content of Ti_3_SiC_2_ reached 58 vol.% [19]. When the Al-Si alloy was employed to infiltrate into TiC preform, the volume content of Ti_3_SiC_2_ reached 52 vol.% [20]. The volume content of Ti_3_SiC_2_ in RMI-based composites is usually lower than 60 vol.% due to the formation of byproducts including SiC, which limits the improvement of EMI shielding performance.

The formation of Ti_3_SiC_2_ in the LSI process includes three steps: the infiltration of liquid silicon, the formation of TiC twins, and the transformation from TiC twins to Ti_3_SiC_2_ [18]. In order to increase the volume content of Ti_3_SiC_2_, we tried to shift the transition from TiC to TiC twins before LSI by introducing Al as a catalyst into TiC preform, and, thus, promoted the transformation from TiC to Ti_3_SiC_2_. In this work, TiC preforms with the incorporation of Al were first prepared, and then LSI was carried out to synthesize Ti_3_SiC_2_-based ceramics. The microstructure and EMI shielding performance of as-obtained ceramics were studied systematically.

## 2. Experimental

### 2.1. Materials Preparation

TiC (HWRK Co. Ltd., >99% purity, Beijing, China) with an average particle size of 1.5 μm, Al (ST-NANO Co. Ltd., >99% purity, Shanghai, China) with an average particle size of 1 μm, and Si with an average particle size of 45 μm (Jinan Yinfeng Silicon Products Co. Ltd., >99% purity, Jinan, China) were employed in this work.

First, TiC and Al powders with different weight fractions were mixed homogenously. Second, the mixed powders were put into a metal mold with dimensions of 75 mm × 15 mm, and a uniaxial pressure of 17 MPa was put on the metal mold, obtaining green TiC preforms. Third, the green TiC preforms were laid in Al_2_O_3_ crucible with SiC powder bed, and were heat-treated at 1400 °C for 1 h with flowing argon. At last, the TiC preforms were infiltrated with liquid silicon at 1600 °C in a vacuum furnace.

The different samples were designated as samples TA3, TA6, TA9, and TA12, according to the different weight ratios of Al in the green TiC preforms (3, 6, 9, 12 wt.%).

### 2.2. Characterization 

The as-fabricated samples were crushed into powders, and an X-ray diffractometer (XRD, Rigaku D/max-2400, Tokyo, Japan) with CuKα radiation was employed to analyze the phase composition. The voltage was 40 kV and the current was 100 mA, and data was recorded in the angle (2θ) ranging from 5 to 80° with a scanning rate of 5 °/min.

The morphology of polished surface and fracture surface was characterized by a scanning electron microscope (SEM, S-4700, Hitachi, Tokyo, Japan) at 15 kV and 10 mA. The back scattered electron (BSE) imaging was employed to characterize the phase distribution, according to the different atomic number contrast, and the secondary electron imaging was employed to observe the fracture surface. An energy dispersive spectrometer (EDS, Genesis XM 2000, EDAX Inc., Berwyn, PA, USA) was used to analyze the element. A transmission electron microscope (TEM, Talos F200X, FEI, Hillsboro, OR, USA) was employed to analyze the microstructure on an atomic scale. 

For EMI shielding tests, the bars with dimensions of 22.86 × 10.16 × 3.00 mm^3^ were cut from the as-fabricated samples, and then the scattering parameters (S-parameters: S_11_, S_12_, S_21_, S_22_) in the frequency range from 8.2 to 12.4 GHz were measured with a vector network analyzer (VNA, MS4644A, Anritsu, Kanagawa, Japan) using the waveguide method, according to ASTM D5568-08. And then the reflected (R) and transmitted (T) coefficients were calculated by the following equations [22].
(1)$$R={\left|S_{11}\right|}^{2}={\left|S_{22}\right|}^{2}$$
(2)$$T={\left|S_{12}\right|}^{2}={\left|S_{21}\right|}^{2}$$

After that, reflection loss (SE_R_), absorption loss (SE_A_), and total shielding effectiveness (SE_T_) were calculated by Equation [22].
SER = −10 × log10(1 − R)(3)
(4)$${\mathrm{SE}}_{A}=-10\times \log_{10}\left(\frac{T}{1-R}\right)$$
(5)$${\mathrm{SE}}_{T}={\mathrm{SE}}_{A}+{\mathrm{SE}}_{R}$$

## 3. Results and Discussion

### 3.1. Phase Composition and Microstructure

After heat-treatment and LSI, the density and open porosity of different samples are listed in Table 1. As the content of Al increased in the TiC preform, the Al melted during the heat-treatment and filled a part of the pores, which results in the change of the density and open porosity. During LSI, the liquid silicon infiltrated into TiC preform with capillary force and reacted with TiC particles. The density of all four samples increased to more than 3.7 g/cm^3^. It noted that the porosity of samples TA9 and TA12 increases slightly compared with that of samples TA3 and TA6. As reported, the transformation from TiC to Ti_3_SiC_2_ led to a volume decrease of 11.7% [23]. Therefore, the variation of porosity may be related to volume shrinkage between the reaction of TiC and liquid silicon.

XRD patterns of as-fabricated samples are shown in Figure 1. TiC particles preferred to react with liquid silicon to form TiSi_2_ and SiC, as reported in the previous work [18]. With the addition of Al, the diffraction peaks of Ti_3_SiC_2_ appear in the XRD patterns of as-obtained ceramics. Samples TA3, TA6, TA9, and TA12 have the same composition as Ti_3_SiC_2_, TiSi_2_, and SiC. With the addition of Al in the TiC preform, it can effectively decrease the formation energy of TiC twins [24,25], which promotes the formation of Ti_3_SiC_2_ in the LSI process.

Figure 2a shows the low-resolution TEM image of Ti_3_SiC_2_-based ceramics in which Ti_3_SiC_2_ and TiSi_2_ grains can be clearly seen. As shown in the high-resolution TEM image (Figure 2b), the periodic and stacking structure of Ti_3_SiC_2_ can be clearly seen, and there is no crystallization relationship between TiSi_2_ and Ti_3_SiC_2_. The typical selected area electron diffraction patterns of Ti_3_SiC_2_ and TiSi_2_ are displayed in Figure 2c, and the corresponding incident beam directions are parallel to [1$\bar{1}$00] and [010] for Ti_3_SiC_2_ and TiSi_2_, respectively. The super lattice structure can be found due to periodically stacking Ti6C and silicon.

As shown in the high-angle annular dark field (HADDF) image (Figure 3a), the laminated structure of Ti_3_SiC_2_ can be clearly seen, and there are two kinds of particles inserting in Ti_3_SiC_2_ grain. By the EDS mapping image, it can be deduced that the left one is Al, and the other is SiC. As shown in Figure 3b, the TiSi_2_ and SiC can be clearly seen, and Al particles distribute as a discontinuous phase. The SiC was formed by the reaction during LSI, and the Al was formed by the condensation of residual Al melt. However, it cannot be detected by XRD, which indicates that the volume content of Al is lower than 5 vol.%. In the LSI process, part of Al dissolved into the Ti_3_SiC_2_ grains, and the others would be condensed to form Al particles.

Figure 4 shows the BSE images of samples TA3, TA6, TA9, and TA12, in which the bright phase and grey phase represent the Ti_3_SiC_2_ and TiSi_2_, respectively. The content of each phase in all four samples were obtained by measuring the areas in the BSE images, and at least 10 images were employed. The SiC and Al have the similar average atomic number, which shows the same contrast. Thus, it is hard to distinguish these two phases by BSE. Since the Al only occupies a very small amount and it is hard to distinguished from SiC, the SiC (Al) in Figure 5 represents the total volume content of SiC and Al. With the addition of Al into the TiC preform, the Ti_3_SiC_2_ and TiSi_2_ are the main phase, and the volume content of Ti_3_SiC_2_ is greatly promoted with the increase of Al content. As shown in Figure 5, the volume content of Ti_3_SiC_2_ increases from 42 vol.% to 85 vol.%, and then decreases to 68 vol.%. With the appearance of Ti_3_SiC_2_, the volume content of TiSi_2_ and SiC decrease, which indicates that a large part of Ti and carbon was consumed to form Ti_3_SiC_2_, and a smaller source of Ti and carbon was consumed to form TiSi_2_ and SiC.

It is interesting to note that no silicon can be detected in all four samples. The similar phenomenon can be found for the infiltration of liquid silicon into TiC-C preform. Once Ti_3_SiC_2_ appeared in the final product, the silicon would disappear. Generally speaking, it is normal to detect the residual silicon in the LSI-based materials. In the LSI process, the silicon infiltrated into the porous preform, and filled into the pores. Part of silicon would be consumed by the reaction, and the others would remain. For high-temperature structural materials like C/C-SiC, the silicon always remained after the infiltration of liquid silicon into porous C/C, which is harmful to the high-temperature strength [26]. However, the different phenomenon can be found for Ti_3_SiC_2_-based ceramics fabricated by LSI, which indicates that the infiltration of liquid silicon was inhibited by the formation of Ti_3_SiC_2_ [23].

### 3.2. Growth Mechanism

Table 2 shows the summary of phase content of MAX phases in RMI-based ceramics. For the infiltration of Al melt into TiC-TiO_2_ preform, the volume content of Ti_3_AlC_2_ reached 40 vol.% [17]. For the infiltration of liquid silicon into TiC-C preform, the volume content of Ti_3_SiC_2_ reached 58 vol.% [19]. For the infiltration of Al-Si alloy into TiC preform, the volume content of Ti_3_SiC_2_ reached 52 vol.% [20]. For the infiltration of Al-Si alloy into TiC-TiO_2_ preform, the volume content of Ti_3_SiC_2_ reached 44 vol.% [21]. As listed in Table 2, the volume content of Ti_3_SiC_2_ in this work can reach 85 vol.%, which is much higher than other works. The catalytic role of Al to the formation of Ti_3_SiC_2_ has been demonstrated, and the difference is the introduction of Al into the TiC preform before LSI.

In the LSI process, the TiSi_2_ and SiC was formed by the reaction of TiC and Si. At the beginning, Equation (6) was used, and then the TiSi_2_ dissolved into liquid silicon to form Ti-Si_rich_ (Equation (7)). As reported, the TiC twins are essential for the precipitation of Ti_3_SiC_2_ [24]. The TiC twins should be formed first (Equation (8)), and then Ti_3_SiC_2_ can be synthesized (Equation (9)).
TiC(s) + 3Si(l) → TiSi_2_(l) + SiC(s)(6)
TiSi_2_(l) + Si(l) → Ti-Sirich(7)
TiC(s) → TiC_twin_(s)(8)
2TiC_twin_(s) + TiSi_2_(l) → Ti_3_SiC_2_(s) + Si(l)(9)

RMI is a reaction-infiltration competition process. When the Al-Si melt was employed, Al lowered the diffusion speed of liquid silicon, and the reaction between liquid silicon and TiC was inhibited. Therefore, TiC remained in the final product [20]. The effect of carbon content on the formation of Ti_3_SiC_2_ has been studied, which revealed that it was essential to form TiC twins by the reaction of carbon with TiSi_2_ [18].

For the infiltration of liquid silicon into TiC-C preform, Equation (6) first started, and then Equation (8) took place. In this work, Al was introduced into the TiC preform before LSI, which may promote the formation of TiC twins before LSI. Equation (8) took place before Equation (6), and, thus, it would increase the transformation efficiency from TiC to Ti_3_SiC_2_, which led to the high-volume content of Ti_3_SiC_2_. Therefore, the Ti_3_SiC_2_ phase content can reach 85 vol.% in this work.

With the consumption of TiSi_2_ in Ti-Si_rich_ melt, the TiSi_2_ preferred to infiltrate inside, and the infiltration of liquid silicon was inhibited. The LSI process was conducted under vacuum, and the liquid silicon was easy to evaporate due to its low vapor pressure. Based on the above analysis, it will be reasonable to understand the disappearance of silicon with the appearance of Ti_3_SiC_2_.

It is noted that the maximum Ti_3_SiC_2_ content can be found for sample TA9. With the further increase of Al content in the TiC preform, the Ti_3_SiC_2_ content decreased from 85 to 68 vol.%. Al play the catalytic role to promote the formation of TiC twins. When too much Al was introduced into TiC preform, it may agglomerate and inhibit the contact between TiC particles with liquid silicon, which leads to the decrease of Ti_3_SiC_2_ content.

### 3.3. EMI Shielding Performance

The SEs of all four samples with the thickness of 3 mm are shown in Figure 6a. The SE_T_ for samples TA3, TA6, TA9 and TA12 is 26, 31, 39, and 28 dB. All the samples have SE_T_ over 25 dB, which means more than 99% of electromagnetic wave can be shielded. These materials meet the requirements of commercial application. The highest SE can be found for sample TA9 with the most fraction of Ti_3_SiC_2_.

The power balance of all four samples is shown in Figure 6b. With the increase of Al content in the TiC preform, the percentage of reflected power increases from 83% to 96%, and the absorbed power decreases from 17% to 3.6%, which reveals that most of the power was reflected. For samples TA3 and TA6, 82.9% and 82.8% power were reflected, and, above 95%, power was reflected for samples TA9 and TA12. Especially for sample TA9, only 0.01% power can transmit, which reveals the excellent EMI shielding performance. Although the absorption loss is the dominating shielding effectiveness, most of the power was mainly reflected, since the reflection took place before absorption.

The electrical conductivities of all four samples are shown in Figure 7. The electrical conductivities of samples TA3, TA6, TA9, and TA12 are 5.34, 5.91, 8.53, and 5.67 S/cm, respectively. The high electrical conductivity is consistent with the high EMI SE value. All four samples have different phase composition and phase distribution, which leads to the different electrical conductivity.

As reported in the references, the electrical resistivity of Ti_3_SiC_2_ [15], TiSi_2_ [27], and SiC [28] is 22 × 10^−6^, 13–16 × 10^−6^, and 10^3^–10^9^ Ω∙cm. Ti_3_SiC_2_ and TiSi_2_ have high electrical conductivity, while SiC is a typical semi-conductor material. It can be found that the volume contents of high-electrical-conductivity phase (Ti_3_SiC_2_ + TiSi_2_) are 77, 86, 93, and 83 vol.% for samples TA3, TA6, TA9, and TA12. The higher volume content of the (Ti_3_SiC_2_ + TiSi_2_) phase is, the higher the electrical conductivity is. The existence of SiC would impede the electron migration, which inhibits the improvement of electrical conductivity. For sample TA9, it has the lowest SiC content among four samples, so it will be reasonable to have the best electrical conductivity. With the increase of electrical conductivity, sample TA9 exhibits the best EMI shielding performance.

The grain size of Ti_3_SiC_2_ also affects the EMI shielding performance. In the lattice structure of Ti_3_SiC_2_ grain, the edge-sharing Ti_3_C_2_ layers are separated by hexagonal nets of the Si layer. The Ti-d-Ti-d bonding dominates the electronic density of states at the Fermi level, while Si does not contribute significantly at the Fermi level. Therefore, the electrical conductivity of MAX phases perpendicular to the c-axis is higher than the parallel one [29,30]. In future work, the Ti_3_SiC_2_ grain with a long c-axis can be designed to further increase the EMI shielding performance.

## 4. Conclusions

In this work, Ti_3_SiC_2_-based ceramics with high EMI shielding performance were synthesized. The Al can play the catalytic role to promote the formation of TiC twins before LSI, and, thus, more TiC can be transformed into Ti_3_SiC_2_ in the LSI process. The volume content of Ti_3_SiC_2_ increases to 85 vol.% when the weight content of Al in the TiC preform increases to 9 wt.%, and then the volume content of Ti_3_SiC_2_ decreases with the further rise of Al content. The EMI shielding effectiveness of Ti_3_SiC_2_-based ceramics can reach 39 dB, revealing good EMI shielding performance.

## Figures and Tables

**Figure 1 materials-13-00328-f001:**
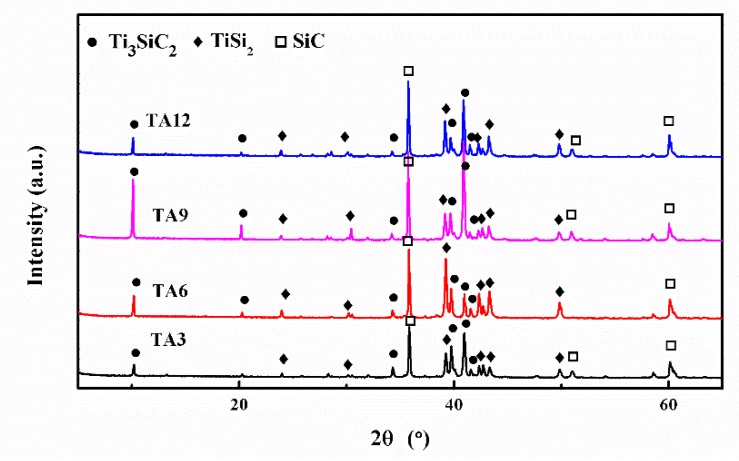
XRD patterns of samples TA3, TA6, TA9, and TA12.

**Figure 2 materials-13-00328-f002:**
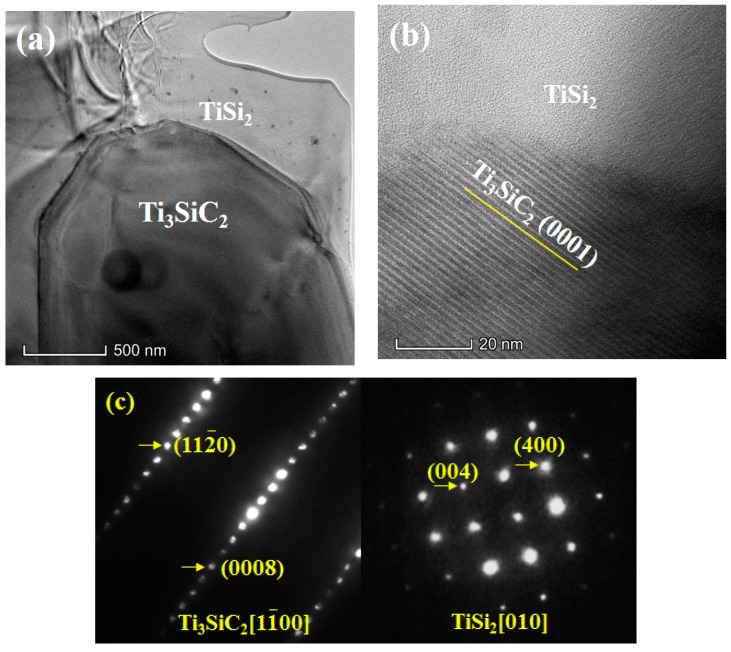
(**a**) Low-magnification and (**b**) high-resolution TEM images showing Ti_3_SiC_2_ and TiSi_2_, (**c**) and the corresponding selected area electron diffraction patterns.

**Figure 3 materials-13-00328-f003:**
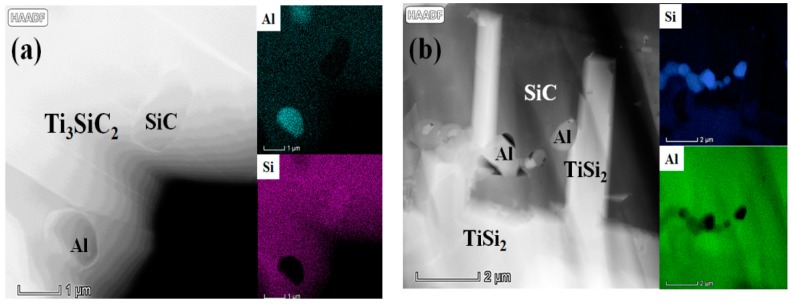
HAADF and EDS imaging images (**a**) showing Ti_3_SiC_2_, Al, and SiC and (**b**) showing TiSi_2_, Al, and SiC.

**Figure 4 materials-13-00328-f004:**
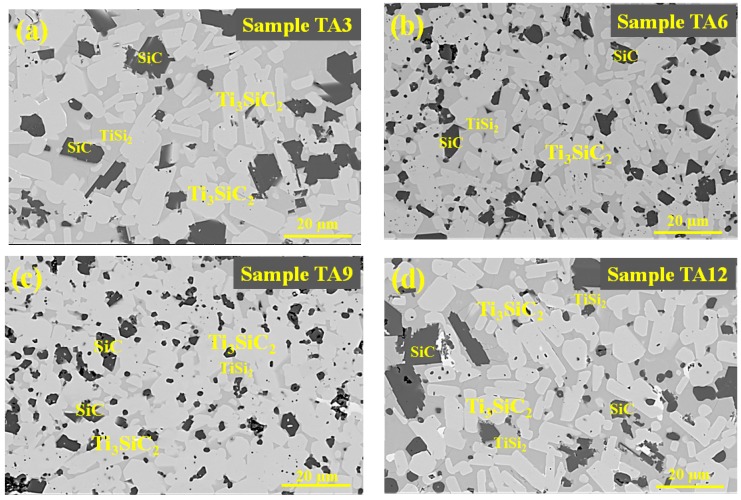
BSE images of samples (**a**) TA3, (**b**) TA6, (**c**) TA9, and (**d**) TA12.

**Figure 5 materials-13-00328-f005:**
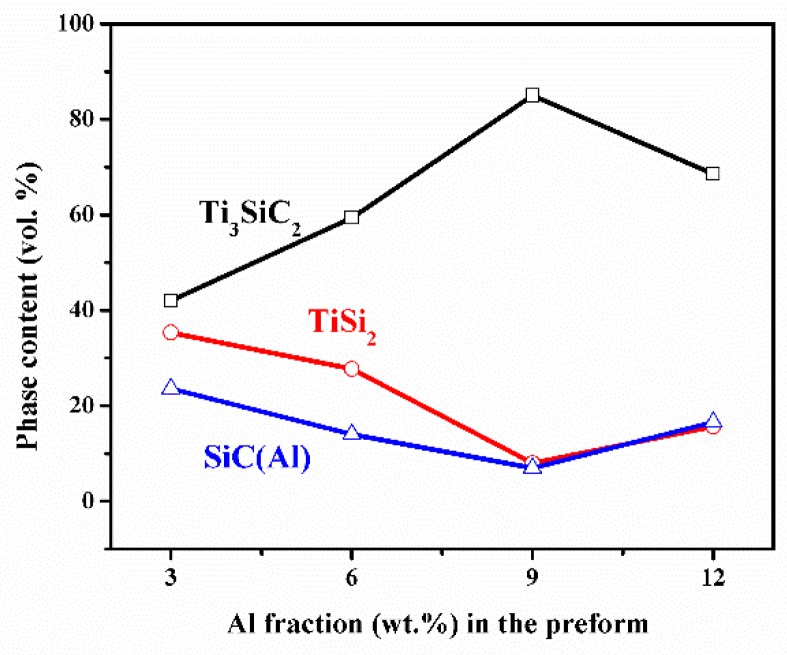
Volume content of each phase calculated from BSE images.

**Figure 6 materials-13-00328-f006:**
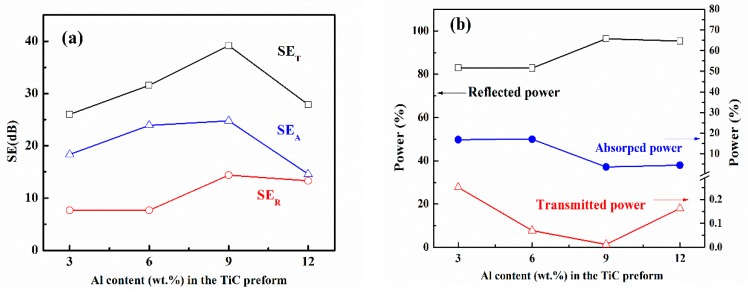
(**a**) SE_T_, SE_R_, and SE_A_ of all four samples with thickness of 3 mm, and (**b**) power balance as a function of Al content.

**Figure 7 materials-13-00328-f007:**
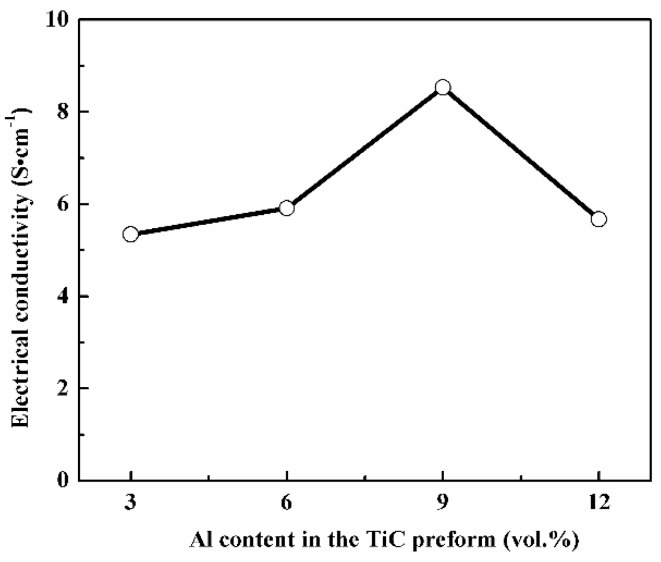
Electrical conductivity as a function of Al content in TiC preform.

**Table 1 materials-13-00328-t001:** Density and open porosity of all four samples before and after LSI.

Processing	Parameters	TA3	TA6	TA9	TA12
After heat-treatment	Density (g/cm^3^)	2.18	2.29	2.26	2.36
	Porosity (vol.%)	54	47	44	43
After LSI	Density (g/cm^3^)	3.8	3.9	3.8	3.9
	Porosity (vol.%)	3	1	5	5

**Table 2 materials-13-00328-t002:** Summary of volume content of MAX phases in the RMI-based ceramics.

Preform	Melt	Phase Composition	Volume Content of MAX Phases (vol.%)	Ref
TiC-TiO_2_	Al	Ti_3_AlC_2_-TiAl_3_-Al_2_O_3_-Al	40	17
TiC-C	Si	Ti_3_SiC_2_-TiSi_2_-SiC	58	19
TiC	Al-Si	Ti_3_SiC_2_-TiAl_x_Si_y_-TiC-Al-SiC	21−52	20
TiC-TiO_2_	Al-Si	Ti_3_SiC_2_-TiC-Al_2_O_3_-Al	44	21
TiC-Al	Si	Ti_3_SiC_2_-TiSi_2_-SiC-Al	42−85	This work

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
