# Peer review of "Synthesis and Electromagnetic Interference Shielding Performance of Ti3SiC2-Based Ceramics Fabricated by Liquid Silicon Infiltration"

_materials, 2020, doi:10.3390/ma13020328_

Round 1

Reviewer 1 Report

The paper is interesting by the use of aluminum as catalyst for twin formation.

1. An explanation of increase of the porosity of the samples TA9 and TA12 after LSI process would be necessary.

2. The discussion of the role of Al would be better supported if micrographs of the green samples would be presented.

3. The first sentences of "Abstract" matches better to "Introduction" section

4. The main problem of the paper is the deficient English. I give a few examples that might create confusions:

   a) Lines 12-13: "...with incorporating small amounts of ..." it is not clear from this sentence if Al is incorporated in TiC prior LSI. in addition, the gerund "incorporating" is less appropriate than the noun "incorporation".

   b) line 23: In this sentence "Electromagnetic interference" appears as grammatical subject  and "attention" as a complement, whereas the position has to be changed (Attention is paid to EM...)

   c) generally, the use of the grammatical article seems to be a problem.

Author Response

1. An explanation of increase of the porosity of the samples TA9 and TA12 after LSI process would be necessary.

    Response: Thanks for your suggestion.

2. It’ noted that the porosity of samples TA9 and TA12 increases compared with that of samples TA3 and TA6. As reported, the transformation from TiC to Ti3SiC2 led to a volume decrease of 11.7%. So the variation of porosity may be related to volume shrinkage between the reaction of TiC and liquid silicon.

3. The discussion of the role of Al would be better supported if micrographs of the green samples would be presented.

    Response: It’s a good suggestion. The emphasis of the present work is to compare the effect of Al on the formation of Ti3SiC2 in the liquid silicon infiltration, and the microstructure difference of as-obtained ceramics was discussed in detailed. By the comparison between the microstructure after liquid silicon infiltration, the role of Al can be clearly revealed. So, it is not necessary to present the micrographs of the green samples.

4. The first sentences of "Abstract" matches better to "Introduction" section

    Response: Thanks for your suggestion. The “Abstract” has been revised.

The main problem of the paper is the deficient English. I give a few examples that might create confusions:

a) Lines 12-13: "...with incorporating small amounts of ..." it is not clear from this sentence if Al is incorporated in TiC prior LSI. In addition, the gerund "incorporating" is less appropriate than the noun "incorporation".

    Response: Thanks for your suggestion. It has been revised in the submitted manuscript.

b) Line 23: In this sentence "Electromagnetic interference" appears as grammatical subject and "attention" as a complement, whereas the position has to be changed (Attention is paid to EM...)

    Response: Thanks for your suggestion. It has been revised in the submitted manuscript.

c) Generally, the use of the grammatical article seems to be a problem.

    Response: Thanks for your suggestion. The English spelling and grammar had been carefully checked before it was submitted to the magazine at this time.

Reviewer 2 Report

The work is about sythesis of Ti3SiC2 MAX phase using liquid silicon infiltration method with additions of Al as catalyst. The manuscript is well written and the results are clearly presented.

I only have a few small comments and questions:

1.) l 89-95, eq. (1)-(5): Can you give a reference for these equations?

2.) l 140-142: Which phase content can be expected from phase diagram data, e.g. with the increasing Al content? Do you have an explanation for the maximum Ti3SiC2 content of sample TA9 in comparison to TA12?

3.) l 148, figure 4: I suggest to add the sample name into the top right corner of each subfigure a)-d).

Author Response

1.) l 89-95, eq. (1)-(5): Can you give a reference for these equations?

Response: Thanks for your suggestion. The references have been added in the submitted manuscript.

2.) l 140-142: Which phase content can be expected from phase diagram data, e.g. with the increasing Al content? Do you have an explanation for the maximum Ti3SiC2 content of sample TA9 in comparison to TA12?

Response: (a) Actually, the phase content can’t be expected from the phase diagram data. As we know, the TiC inclined to react with liquid silicon to form TiSi2 and SiC. Only with the existence of Al as the catalyst, Ti3SiC2 can be formed. Without Al, no Ti3SiC2 can be formed. So it can be deduced that the phase composition depends on the phase content of Al, and it is hard to expect the phase content by the phase diagram data.

(b) The explanation for the maximum Ti3SiC2 content of samples TA9 are added as follows.

It’s noted that the maximum Ti3SiC2 content can be found for sample TA9. With the further increase of Al content in the TiC preform, the Ti3SiC2 content decreased from 85 to 68 vol.%. Al play the catalytic role to promote the formation of TiC twins. When too much Al was introduced into TiC preform, it may agglomerate and inhibited the contact between TiC particles with liquid silicon, leading to the decrease of Ti3SiC2 content.

3.) l 148, figure 4: I suggest to add the sample name into the top right corner of each subfigure a)-d).

Response: Good point. The figures have been revised in the submitted manuscript.